# A Technology-Based Intervention to Support Older Adults in Living Independently: Protocol for a Cross-National Feasibility Pilot

**DOI:** 10.3390/ijerph192416604

**Published:** 2022-12-10

**Authors:** Vera Stara, Margherita Rampioni, Adrian Alexandru Moșoi, Dominic M. Kristaly, Sorin-Aurel Moraru, Lucia Paciaroni, Susy Paolini, Alessandra Raccichini, Elisa Felici, Lorena Rossi, Cristian Vizitiu, Alexandru Nistorescu, Mihaela Marin, Gabriella Tónay, András Tóth, Tamás Pilissy, Gábor Fazekas

**Affiliations:** 1Model of Care and New Technologies, IRCCS INRCA-National Institute of Health and Science on Aging, Via Santa Margherita 5, 60124 Ancona, Italy,; 2Department of Psychology, Education and Teacher Training, Transilvania University of Brasov, B-dul Eroilor 29, 500036 Brasov, Romania; 3Department of Automatics and Information Technology, Transilvania University of Brasov, B-dul Eroilor 29, 500036 Brasov, Romania; 4Neurology Department, IRCCS INRCA-National Institute of Health and Science on Aging, Via della Montagnola 81, 60100 Ancona, Italy; 5Institute of Space Science, Atomistilor Str. 409, 077125 Magurele, Romania; 6National Institute of Locomotor Diseases and Disabilities, National Institute for Medical Rehabilitation, Szanatórium utca 19, 1121 Budapest, Hungary; 7Department of Manufacturing Science and Engineering, Faculty of Mechanical Engineering, Budapest University of Technology and Economics, Muegyetem rkp 3., 1111 Budapest, Hungary; 8Department of Rehabilitation Medicine, University of Szeged, Dugonics Square 13, 6720 Szeged, Hungary

**Keywords:** active ageing, older adults, technology-based intervention, protocol

## Abstract

Innovative technologies can support older adults with or without disabilities, allowing them to live independently in their environment whilst monitoring their health and safety conditions and thereby reducing the significant burden on caregivers, whether family or professional. This paper discusses the design of a study protocol to evaluate the acceptance, usability, and efficiency of the SAVE system, a custom-developed information technology-based elderly care system. The study will involve older adults (aged 65 or older), professional and lay caregivers, and care service decision-makers representing all types of users in a care service scenario. The SAVE environmental sensors, smartwatches, smartphones, and Web service application will be evaluated in people’s homes situated in Romania, Italy, and Hungary with a total of 165 users of the three types (cares, elderly, and admin). The study design follows the mixed method approach, using standardized tests and questionnaires with open-ended questions and logging all the data for evaluation. The trial is registered to the platform ClinicalTrials.gov with the registration number NCT05626556. This protocol not only guides the participating countries but can be a feasibility protocol suitable for evaluating the usability and quality of similar systems.

## 1. Introduction

In 2050, there will be 1.5 billion people aged 65 worldwide [1]. The central policy in response to this global trend is to foster the concept of Active and Healthy Ageing (AHA) [2,3]. Despite the influence of genetics, the health of the older population is associated mainly with the physical and social environments of their daily life, such as homes, neighborhoods, and communities, and of their individual characteristics, such as gender, ethnicity, and socioeconomic status [4,5].

Innovation technologies can support older adults, with or without disabilities, by enabling an independent life in their environment and monitoring their health status and safety, therefore reducing the significant burden of care for caregivers, whether family and/or professional [6,7,8]. Using smart artifacts can also be an effective way to overcome social isolation through connecting with the outside world, obtaining social support, engaging in activities of interest and strengthening self-confidence [9,10,11,12]. In addition, innovative technologies can improve the safety of older people through emergency solutions (e.g., fall detection, alert function, help call opportunities) by supporting therapy, observing physical parameters, and controlling the environmental ones [13,14,15,16,17]. Khosravi et al. [18] reported that key issues in aging care are the risk of falls, chronic diseases, dementia, social isolation, depression, poor well-being, and insufficient medication management. He raises the point that assistive technologies can improve the quality of life, especially among older adults, by tackling each of the issues previously outlined.

Although there is limited understanding of the digital landscape for aging care [19], several Smart Home technologies (e.g., web platforms, applications, sensors) that support older adults’ quality of life, security, and growing sense of loneliness have been developed in recent years. The success of these technologies still depends on users’ perception of personal privacy [20], so data security and confidentiality are a priority for the acceptance of a Smart Home system. To address these challenges, several projects have been developed. For example, after a 12-week Smart Home personalized technology program, participants’ quality of life increased significantly [21], the Fik@ room web platform allowed social connection among older adults in a secure digital environment [22], and the SMART4MD mobile application [23] facilitated a sense of coherence for older persons with cognitive impairment.

The COVID-19 pandemic has had an enormous impact on the extent of social connectedness and the quality of relationships among individuals, and it will probably represent a general rethinking of new methods and (digital) models of care [19]. Different approaches for the involvement of older people in various activities and reducing social isolation have been proposed in the last two years [24]. The Pharaon Project [25] addressed the importance of alternative approaches to face-to-face methods using phone and online interviews, online questionnaires, and virtual and semi-virtual co-creation seminars. Moreover, telemedicine and e-Health services have proven essential to support care for older adults and family caregivers. For example, Internet-based technology has helped people at early stages of cognitive impairment through electronic reminders, daily activities, and cognitive stimulation therapy and games [26].

Based on this scenario, eight partners from three countries (Italy, Hungary, and Romania) joined their efforts in the European-funded SAVE (SAfety of elderly people and Vicinity Ensuring) project (EU Grant Agreement AAL-CP-2018-5-149). The SAVE system aims to offer technology-based support to older adults to stay in their familiar surroundings for as long as possible while feeling safe and optimally cared for. The SAVE technology has been designed and developed according to the User Centered Design (UCD) approach, which involves multiple interactions with users to understand their needs and preferences and involve them in the design process for creating a helpful and appreciated technological product [27]. Secondarily, it supports informal caregivers, such as relatives, in providing optimal care for their loved ones while maintaining their professional and private life.

### Study Objectives

The general objective of this study is to test the usability and efficiency of the SAVE prototype by systematically and objectively identifying the strengths and weaknesses of the proposed solution for enabling older adults to keep their independent and active lives in their homes and maintain their social relationships for as long as possible.

## 2. Methodology

### 2.1. Study Design

To thoroughly evaluate the SAVE prototype, a pre-post interventional study involving the use of the SAVE platform for 21 consecutive days was designed. It consists of the use of a mixed-methods approach, in which it collects both qualitative (open questions) and quantitative (standardized tests) data in three different measurements (T0, T1, T2) during the period of use of the system. The data collection card will therefore be divided into three different sections, which correspond to the three different moments of detection:Prior to the start of the experimentation (T0);Ten days into the intervention study, i.e., at the midterm of the trial (T1);After 21 days, i.e., at the end of the trial (T2).

The users’ data will be logged and continuously stored over the 21-day test period. The research will be managed by qualified personnel, and the researchers will supervise the tests and the interactions between the users and the system.

### 2.2. Study Setting

The study involves three different European institutions in charge of performing the feasibility assessment: the Istituto di Ricerca e Cura a Carattere Scientifico, Istituto Nazionale Ricovero e Cura Anziani (IRCCS INRCA located in the city of Ancona in Italy, the National Institute for Medical Rehabilitation (NIMR) in the city of Budapest in Hungary and the Transilvania University of Brașov (UNITBV)in collaboration with the Romanian Direction of Social Assistance (DAS) located in the Brasov City Council and the Timișoara City Council and with ”Hand in Hand” Association from Brașov in Romania. This multicentric setting will allow for assessing the SAVE system in different social and cultural contexts. Overall, the cross-national approach will ensure a broad acceptance of the developed technology and prepare the possibility of its dissemination and the transferability of the research methods adopted by the SAVE study at the European level and well beyond the initial life cycle of the project.

### 2.3. Participants

The study will involve primary, secondary and tertiary users. According to the European Project SAVE activities, the primary users are expected to be 80 older adults: 30 participants will be enrolled in Romania, 25 in Hungary and the remaining 25 in Italy. Their caregivers (30 participants will be enrolled in Romania, 25 in Hungary and 25 in Italy) as secondary users and at least 5 tertiary users for the site will be enrolled.

Inclusion and exclusion criteria for each end user group are described in Table 1.

### 2.4. Recruitment

Potential participants will be selected based on free participation and effective adherence to the inclusion criteria. The staff involved in the study will contact their networks (associations, recreational centers, trade unions, etc.) by phone and/or email, which will provide the names of potential participants. The latter will be contacted by phone to verify the inclusion criteria and describe the objectives, methods, procedures and timing of the study. Once compliance with the inclusion and exclusion criteria of the study is verified and informed consent is obtained, the local team member will proceed with the baseline evaluation. 

### 2.5. Trial status

User recruitment was started in (enter date here), which was completed by (enter method here). The study is currently underway, with data collection beginning in September 2022 and expected to end by March 2023.

### 2.6. The Intervention

The SAVE system will be implemented in end users’ homes, which is achieved by installing all the kits in the appropriate rooms and by offering relevant training for using all the different devices to the end-users.

Flood sensors will be installed in the bathroom (preferably next to the washing machine) and kitchen (preferably next to the dishwasher), presence sensors in the living room and bedroom, and the contact sensor will be installed at the entrance door. The sensors are powered by button batteries with very low energy consumption; the producer of the sensors advertises an autonomy of 2 years with a standard CR2032 button battery. Only the sensors’ hub and the SAVE Sensor Adapter are powered from a socket (the SAVE Sensor Adapter is powered by the USB connector on the sensors’ hub). The other devices can be used as long as they are charged (the smartwatch and smartphone). Thus, the end user’s responsibility is to charge their smartwatch and not to unplug the sensors’ hub (and, in Hungary, the smartphone from the charger). It is planned to place the central unit in the bedroom, where the user can easily charge the smartwatch in the evening. For the best user experience, the required software (for the sensors kit and the smartwatch) will be installed on the users’ smartphones in Romania and Italy. On the Hungarian side, users are provided with a separate mobile phone to receive data and transmit it to the cloud system.

The sensor set and the smartwatch software are only downloaded to the user’s mobile phone at the user’s request. The sensors have been located in places where they do not affect the daily activity of the users, these being easy to move according to preferences and small enough to blend in the background. Thus, at the end of the installation, users will have the Aqara Home System in their homes (5 sensors and a sensor hub), a Samsung smartwatch, and a SAVE Sensors Adapter, all of which are connected to a router with unlimited internet access.

After installing the system, there will be a brief instruction training session on the use and purpose of these devices and the services they offer. This will ensure that the end users are encouraged to use them, performing some tests with them:Heart rate testing in association with the frequency indicated on the smartwatch;Testing the emergency system by pressing the power button 3 times;Testing of the flood and door sensors by visualizing the values received by the SAVE cloud app through the SAVE Web App;Calling a friend/relative from their smartwatch.

To test the pilot solution, the following step-by-step guide will be followed:Kit creation (from the SAVE Admin Centre)—a unique kit key will be generated;Adding devices to the kit (1 x Save Sensor Adapter and 1 x Galaxy Watch 3);Checking the internet connection of the phone/home router in Romania/Italy and the provided mobile phone in Hungary);Verify that the user has a Gmail account; for those users who do not have an account, an account is created;User account creation (self-register—from the SAVE Web App) (Figure 1);Filling in the user profile, including the Kit Key;Installation of the Aqara Home app from the Play Store;Installation and placement of Aqara hub and sensors by following the instruction in the Aqara Home app (Figure 2);Test the functioning of all sensors through the Aqara Home app;Adding a remote control for the Sony projector from the Aqara Home app on the Aqara hub;Adding the SAVE automations in the Aqara Home app by following the instruction in the Aqara Home app and the SAVE installation manual (Figure 3);Installation and configuration of the SAVE Sensor Adapter using WPS or SWS (Figure 4); the SWS is performed by connecting the SAVE Sensors Adapter to a PC/laptop and using the SAVE Sensor Adapter Wi-Fi Setup application desktop application to set the Wi-Fi settings (Figure 5);Test the sensors through the SAVE Platform;Installation of Galaxy Wear app from the Play Store;Pairing the smartwatch to the Galaxy Wear app;Activate Debug mode on the smartwatch;Installation of the two smartwatch apps using the sdb tool provided by Tizen Studio (save-configuration.wgt and save-tizen-watch-face.wgt);Configure the smartwatch for SAVE by filling in the native ID into the SAVE configuration app (Figure 6);Setting the smartwatch face to the SAVE watch face (Figure 7);Configuration of the SOS and fall detection features from the Galaxy Wear app;Test the emergency system. Furthermore, test the data sent from the watch together with the data sent from home (Figure 8);Creation of caregivers’ SAVE Web accounts;Linking the caregivers accounts to the end-user account via the SAVE Web app.

On the Hungarian side, the 7th, 10–21 is completed just before the system is deployed.

### 2.7. The Outcomes

In accordance with the general objectives of this study, primary and secondary outcomes are described in the following subsections.

#### 2.7.1. The Primary Outcomes

The primary outcomes are:Usability, which is understood as “the extent to which a product can be used by certain users to achieve certain goals with effectiveness, efficiency, and satisfaction in a given context of use”. This result will be measured through the SUS scale [28] and the UEQ questionnaire [29].Learnability of the system, which is seen as a component of usability, is the degree to which an interface is intuitive, and the user can immediately understand how to interact with the system. This result will be measured through the SUS scale [28].Acceptance, seen as the degree to which users come to accept and use a piece of technology. This result will be measured through the SUS scale [28] and the UEQ questionnaire [29].

#### 2.7.2. The Secondary Outcomes

The following outcomes will be the secondary effects that we will measure as part of our intervention:Well-being is “a state of complete physical, mental and social well-being, and not simply as the absence of disease”. This result will be measured through the WHO-5 Index [30] and the EQ-5D-5L questionnaire [31].Self-efficacy is the set of beliefs we have about our ability to complete a certain task. This result will be measured through the short version of the GSE self-efficacy scale [32].

### 2.8. Data Collection

In line with the design of the study, three different data collection tools were developed.

As reported in Table 2, the data collection sheet for primary users consists of four dimensions sections, which include a series of scales, as follows:(A)Health and Wellness Condition:Mini-Mental State Examination (MMSE) [33] is a neuropsychological test for the evaluation of disorders of intellectual efficiency and the presence of cognitive impairment. The test consists of 30 questions, which refer to various cognitive areas: orientation in time and space, recording of words, attention and calculation, re-enactment, language, and constructive praxis. The total score is between a minimum of 0 and a maximum of 30 points. A score of 26 to 30 is an indication of cognitive normality. The score is adjusted with the coefficient for age and schooling [34].Functional Ambulation Category (FAC) [35] is a scale that evaluates the ability to achieve autonomy in walking. The ambulatory capacity is evaluated with a score ranging from 0 to 5, where 0 indicates total dependence and 5 indicates complete independence. From the score obtained, it can be deduced the amount of support that the patient requires when walking and on what kind of surfaces he is able to walk.The Barthel Index [36] is an objective and standardized tool for measuring functional status. The individual is scored in a number of areas depending upon the independence of performance. Total scores range from 0 (complete dependence) to 100 (complete independence).SF-12v2 ™ Health Survey [37] is a widely used instrument and is a 12-element subset of the SF-36v2 ™. It is a short and reliable measure of the general state of health. It is useful in health surveys of large populations and has been widely used as a screening tool.Five Well-Being (WHO-5) Index [30] is a short self-reported measure of current mental well-being.EuroQol-5 Dimension-5 Level (EQ-5D-5L) [31] is a self-report survey that measures the quality of life across 5 domains: mobility, self-care, usual activities, pain/discomfort, and anxiety/depression. Each dimension is scored on a 5-level severity ranking that ranges from ‘No problems’ through ‘Extreme problems’.
(B)Self-efficacy:General Self-Efficacy Scale (GSE) [32]. Its abbreviated form of ten entries is a reliable and valid tool for assessing general self-efficacy.
(C)Usability and Acceptance:
System Usability Scale (SUS) [28] is a reliable tool for measuring usability. It consists of a 10-item questionnaire with five response options for respondents from ‘Strongly agree’ to ‘Strongly disagree’. It allows for evaluating various products and services, including hardware, software, mobile devices, websites, and applications. It is easy to administer to participants, can be used on small sample sizes with reliable results, and can effectively differentiate between usable and unusable systems.User Experience Questionnaire (UEQ-S) [29]. This short version of the questionnaire measures the subjective impression of users towards the user experience of products. The UEQ is a semantic differential with 26 items. Both classical usability aspects (efficiency, perspicuity, dependability) and user experience aspects (originality, stimulation) are measured.Quebec User Evaluation of Satisfaction with assistive Technology (QUEST—Version 2.0) [38] is a 12-item outcome measure that assesses user satisfaction with two components, Device and Services.
(D)Privacy and Stigmatization
Open questions

The data collection sheet for secondary users consists of a sequence of 5-Likert scale questions on the following dimensions (Table 2):Usefulness of the system,Reliability of the system,

and some free comments about satisfaction with the system.

The data collection sheet for tertiary users consists of a sequence of 5-Likert scale questions on the following dimensions (Table 2):Impact of the system on the reduction in time spent in caregiving activities;Impact of the system on the reduction in the cost of caregiving activities;Impact of the system on the reduction in the workload.

Each instrument will be administered, if possible, in a face-to-face session in the presence, otherwise remotely, of a trained interviewer, who will report the answers on a paper version of the data collection card.

During use, the following log data will be continuously recorded: usage time/used services (time/number), number of interactions, participation in social events (web conference, phone call, video call), tracked daily activities, number of errors, number of aids required for the tasks, recognized user commands, used input modality, captured touch screen data/activity, number of services used in the given period, number of tasks solved by using the SAVE system (for example, did the user recieve an answer to a question, did the user manage to call/inform the caregiver, was the user able to perform recreational activities with the help of the device, did the user manage to send the alarm to the caregiver, whether the caregiver was informed about the user’s fell, could not sleep, decreased activity, be deteriorated, etc.).

### 2.9. Data Analysis

The proposed study is a small-scale study, carried out as part of an innovation and research project. To calculate sample size, G*Power software (latest ver. 3.1.9.7; Heinrich-Heine-Universität Düsseldorf, Düsseldorf, Germany) [39] was used. The G*Power software supports sample size and power calculation for various statistical methods, including dependent t-test (differences between two dependents means—matched pairs). A power calculation based on a dependent t-test shows that at least 34 end users per group are needed (alpha set at 0.05, beta 0.2 and effect size at 0.5 medium). It will conduct only a partial evaluation of the effectiveness of the services to specific dimensions of the quality of life of the subjects involved. To investigate primary and secondary objectives, the data will be analyzed using qualitative and quantitative methods and mixed analysis methods.

The processing of questionnaire data will be performed through specific software, such as SPSS, Mplus (for the execution of the confirmatory factor analysis), and AMOS (for the execution of the exploratory factor analysis), depending on the needs. The quality of the data and its internal consistency will be evaluated using the Cronbach Alpha and other specific tests. The questionnaires will first be verified manually to check the completeness of the compilation and any apparent inconsistencies. Later, automated routines will be used to detect dubious outliers and records. In such cases, the necessary data cleaning will be carried out.

An Analysis Plan will be defined on the basis of which the analyses themselves will then be conducted. The first step of the analysis will be exploratory in nature. The descriptive analysis of the sample will be conducted through the classic techniques of uni- and bi-variate statistical analysis. Significant differences between outcomes and exposures will be compared using the Chi Quadro test, the Fisher Exact, the t-test, or the Anova test. The characteristics of the subjects will be compared with those of the non-respondents to verify any distortions due to non-responses during the detection.

## 3. Discussion

In this study, we have described a feasibility protocol to evaluate the usability and acceptability of the SAVE system for enabling older adults to keep their independent and active lives in their homes and to maintain their social relationships for as long as possible. This is closely related to the concept of “aging in place”, which represents the desire expressed by older people to continue living within the community, with some level of independence, rather than in residential care [40]. Housing and neighborhood satisfaction have been used as good indicators of environmental and overall well-being for older adults [41]. Technological devices should provide a sense of confidence and security for older adults, which would enable aging in place. In that sense, the SAVE intervention has the potential to assist older people to live in their own homes and facilitate their engagement in everyday tasks, helping to maintain their independence and increasing their control over the world around them. Several studies [42,43,44,45,46,47] have claimed that older people want to remain independent for as long as possible. The desire to stay independent stemmed from their wish to not be perceived as a burden to family, friends, or society. Therefore, the smart home solutions would be designed to help older people carry out everyday activities and lead healthier and more fulfilled lives by improving their physical safety and social communication. According to Lee and Kim [48], older adults’ active participation in social activities and establishing their sense of belonging as social members have important effects on successful aging in place. The SAVE system would effectively overcome social isolation among older people by connecting to the outside world, gaining social support, engaging in activities of interest, and boosting self-confidence.

The study must be performed the same way in all participating countries to obtain comparable data from the three countries. However, there are also limitations in the applicability of the intervention. According to Quan-Haase et al. [49], age-related factors beyond income, education, and gender affect older adults, hindering their ability to take advantage of digital technology. Low digital literacy could hinder older people’s use of digital media, perhaps because they did not grow up with it and had to learn and adapt in order to use them later in life [50]. A limited number of participants with a specific level of cognitive impairment is also a limitation, as well as the fact that the recruitment of the subjects and the beginning of the trial will take place when the COVID-19 pandemic is still an ongoing threat. Despite this, in emergency situations, technology has proved to greatly help in mitigating the consequences of physical distancing and helping older people maintain relationships with their relatives and friends [51].

## 4. Ethics and Dissemination

The study was approved by the Ethical Committees of the three countries involved. Any protocol modifications will be notified to the same Ethics Committees and to other interested parties, such as researchers and participants. For the latter, changes will be communicated by email. The principles of the Declaration of Helsinki and Good Clinical Practice guidelines will be adhered to. Participants in this study will provide written informed consent.

### 4.1. Risk Management, Mitigation, and Possible Limitations for the Users

We do not expect any adverse effects on users’ health related to testing the technology platform. The hardware devices used are commercial devices (Samsung smartwatches and AQARA-XIAOMI environmental sensors) and CE-certified. Researchers will provide clear and detailed information on the terms of use of the technology platform and the services offered during the study. The proposed services aim to support older people in terms of personal and residential security by allowing them an independent and safe lifestyle, even when the first signs of disorientation could lead the individual and their family members to progressive isolation and social exclusion, and do not replace (in whole or in part) the support from professional services, as the proposed technologies support the well-being of the participants.

The users who take part in the study will not incur any direct or indirect costs related to the use of the technology platform. All the devices necessary for the trial (smartwatches, smartphones, sensor kits and internet connection) will be made available to users free of charge.

Even if we do not expect any adverse effects, the presence and use of technological devices (sensors and smartwatches) could be a source of discomfort, anxiety about making mistakes, or quickly forgetting the instructions and feeling stigma. Low digital literacy and the poor usability of the devices could hinder older people’s use of technologies. The SAVE technology has been designed and developed according to the UCD approach to counteract these possible limitations by matching the needs and capabilities of users, thereby improving the user experience [27]. The pre-post interventional study design will allow us to better analyze changing behaviors regarding usability, acceptance, and stigma.

The use of a specific manufacturer’s company (Galaxy Gear app) probably significantly limits the range of users, but the operating systems for smartwatches are still very fragmented, and the budget resources of the project are limited. Therefore, for our research, we chose a product of a large company with a significant market share. We aimed for a product with LTE connectivity (based on eSIM), which makes it independent of a smartphone and allows a better monitoring process. At the time of selection, there were not many products with LTE connectivity, and the eSIM technology was not yet supported by all service providers for all smartwatch producers. The architecture we adopted lends itself to the easy inclusion of other devices from other manufacturers by adding a software element (an adapter), which is in our expansion plans.

### 4.2. Data Management

Personal data collected during the trial will be handled and stored following the General Data Protection Regulation (GDPR) 2018. The use of the study data will be controlled by the principal investigator. All data and documentation related to the trial will be stored in accordance with applicable regulatory requirements, and access to data will be restricted to authorized trial personnel. The trial will be run by principal investigators and co-investigators of the three countries involved.

### 4.3. Dissemination

The dissemination program will involve peer-reviewed scientific journals and national and international conferences. The results will be disseminated to all participants.

## 5. Conclusions

The SAVE system was developed using environmental sensors, smartwatches, smartphones, and a Web application to organize and provide improved service levels to older people, their caregivers, and care organizers. The main functions of the SAVE technology are fall detection, activity level monitoring, heart rate monitoring, door opening detection, water leakage detection, localization, enhanced touch and vocal communication, text messaging, and emergency calls. Data are stored in a cloud-based system that is securely accessible to older care recipients, their caregivers, and care managers.

The cross-national study was designed to evaluate the acceptance, usability, and efficiency of the SAVE system among 165 users of the three types. The exclusion and inclusion criteria for each user type were defined. The planned test environment is the elderly user’s home and the caregivers’ and care organizers’ Web platforms through their applications. The convenient user recruiting method will be applied with the help of retirement clubs and referrals through contacts. Training on using smart devices takes place during the first face-to-face meeting at system installation. The task of the older adult is to wear the watch during the day, put it on the charger at night, and test the alarm and message-receiving functions. Caregivers and care organizers are responsible for reacting to any dispatched notification and regularly opening the SAVE interface, reviewing the information on the interface, and using the information found as possible. The use of a control group is not planned.

The study design follows the mixed method approach where standardized tests and questionnaires, open-ended questions, and log data are collected from the users and the cloud database. The described study design could serve as an inspiring study design framework for similar usability and effectiveness studies.

## Figures and Tables

**Figure 1 ijerph-19-16604-f001:**
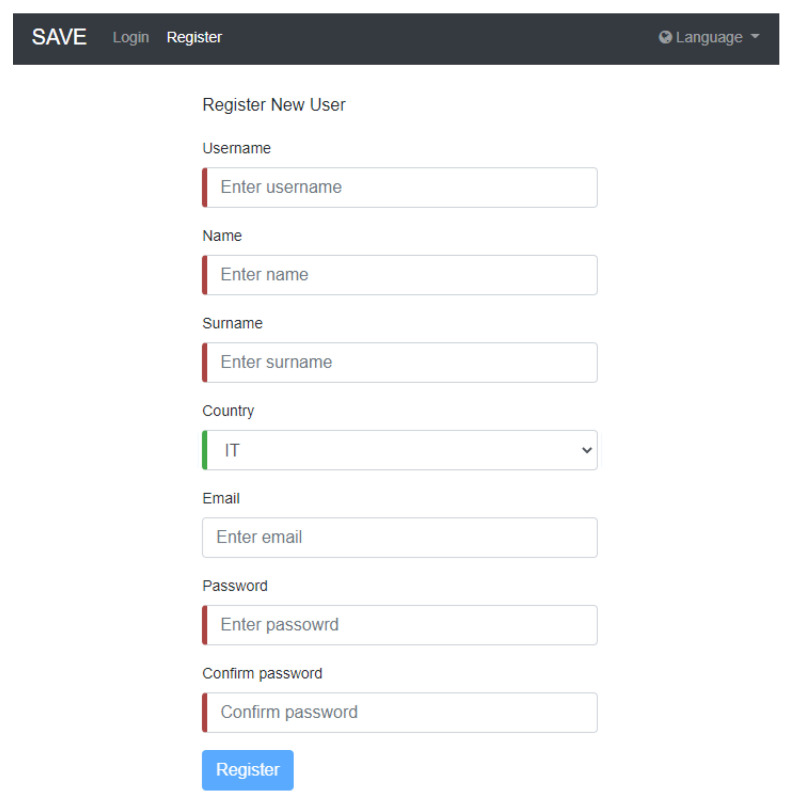
User account creation.

**Figure 2 ijerph-19-16604-f002:**
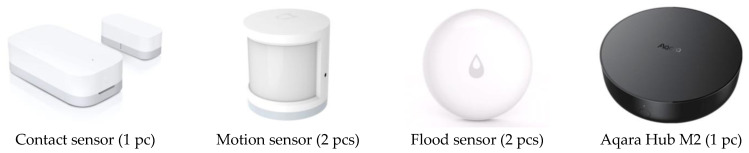
Aqara Home kit.

**Figure 3 ijerph-19-16604-f003:**
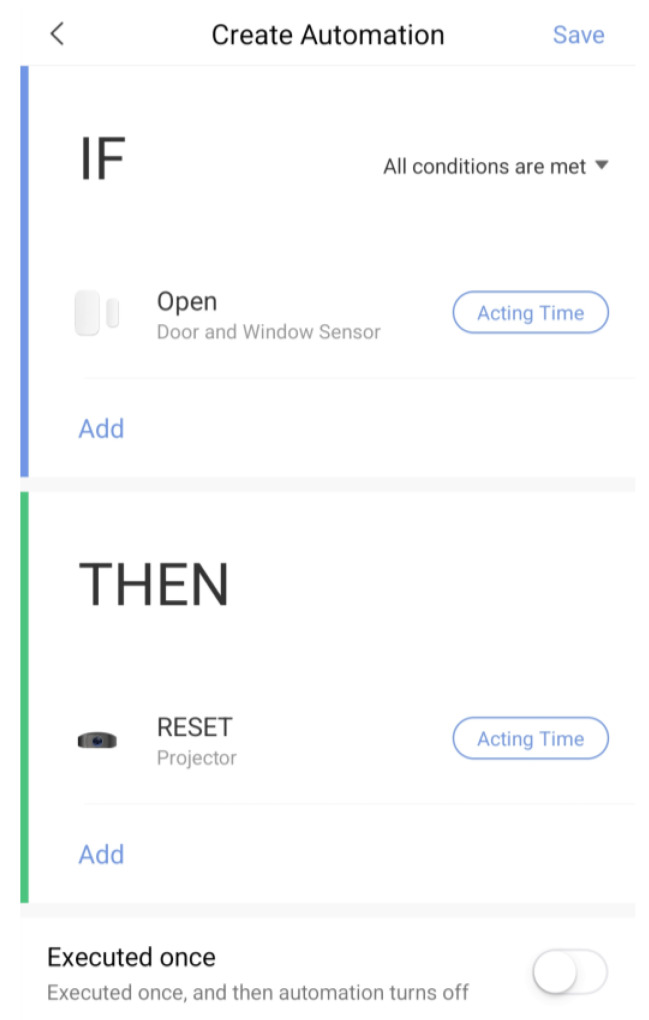
Automation sample.

**Figure 4 ijerph-19-16604-f004:**
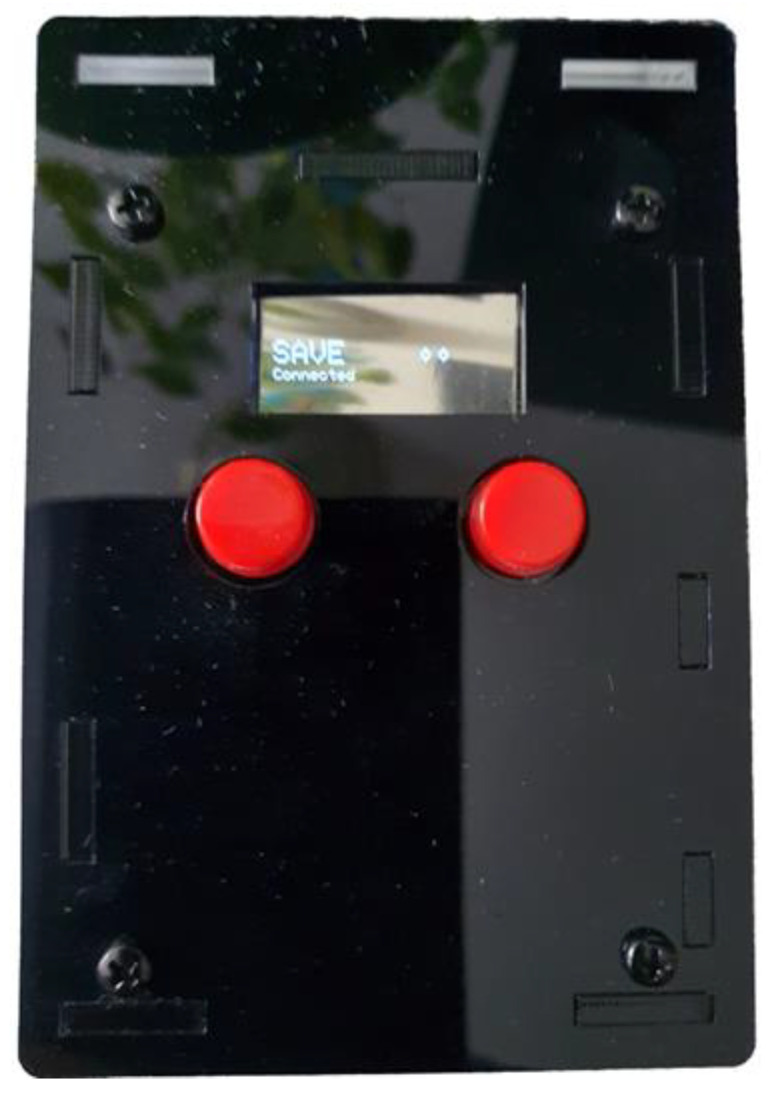
SAVE sensors adapter.

**Figure 5 ijerph-19-16604-f005:**
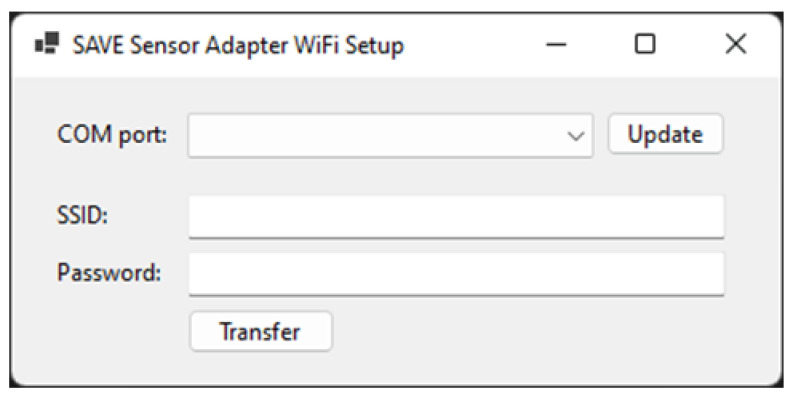
SAVE sensor adapter Wi-Fi setup application.

**Figure 6 ijerph-19-16604-f006:**
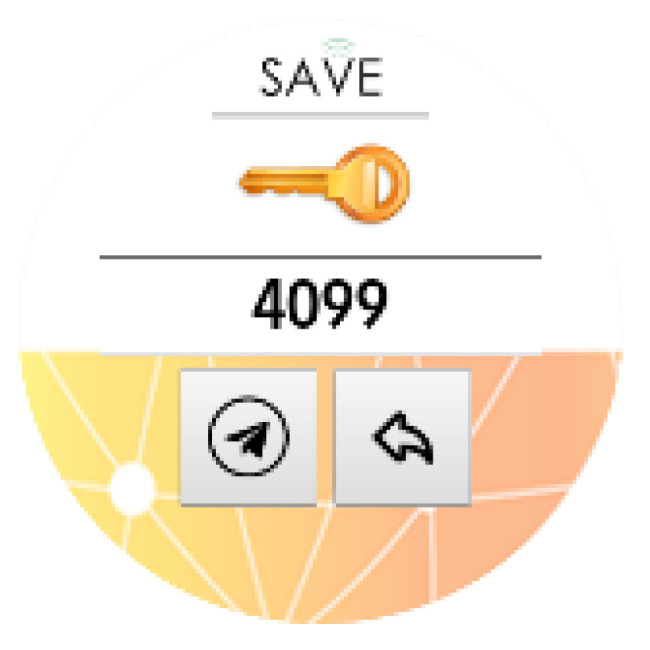
SAVE configuration app.

**Figure 7 ijerph-19-16604-f007:**
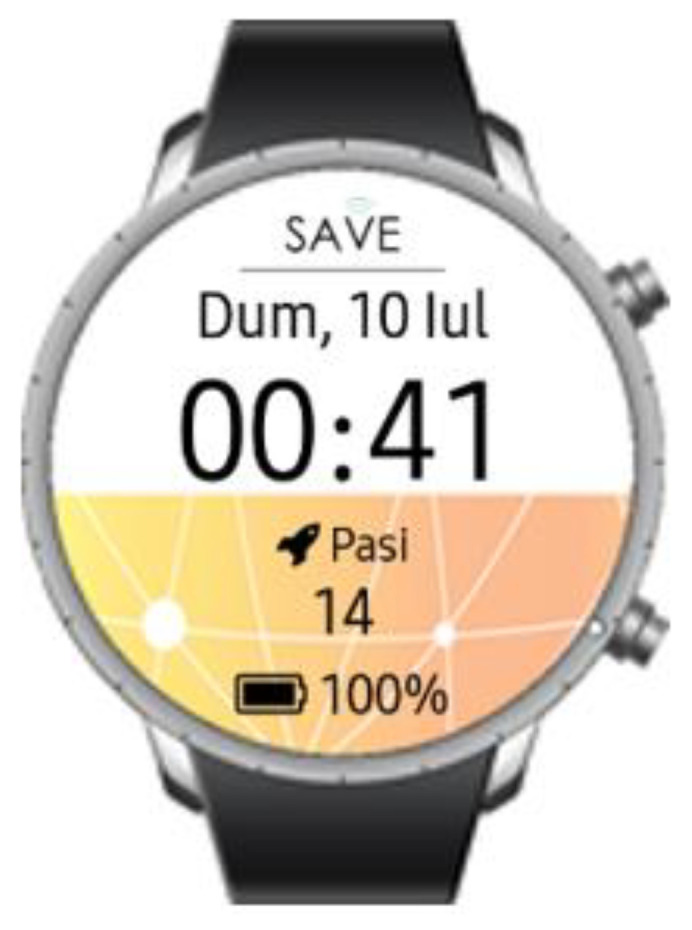
SAVE watch face.

**Figure 8 ijerph-19-16604-f008:**
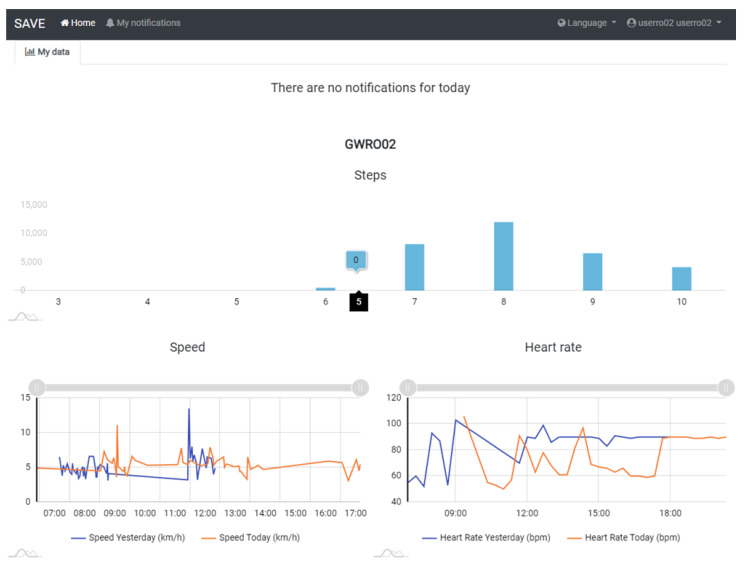
Samples of visualization of user data.

**Table 1 ijerph-19-16604-t001:** Eligibility criteria for primary, secondary, and tertiary users.

End User Type	Inclusion Criteria	Exclusion Criteria
Primary users	- Age ≥ 65 years - Mini Mental State Evaluation (MMSE) between 21 and 27 - Healthy or mild to moderate chronic illness or musculoskeletal disease - Feel physically fit to participate in the study (assessment with FAC and Barthel Index): sufficiently capable of moving, able to maintain and change their position, manipulate and move objects, move in their place of residence, experience the surrounding environment, move by means of transport - Live alone - Interest in the project - Willingness to sign the written informed consent - Able to perform the tasks suggested by the caregiver - Able to use smartphone, smartwatch - Presence of a caregiver	- Age < 65 years - MMSE < 21, subjects diagnosed with dementia or with MMSE ≥ 28 or ≤20 - Participants suffer from severe chronic disease (e.g., symptomatic cardiovascular or respiratory disease, myocardial infarction or stroke in the last 6 months, presence of significant visual and/or auditory impairment, severe metabolic dysfunction, oncological pathologies) or severe disability - No agreement on written informed consent - Participants are carriers of cardiac pacemakers or implantable defibrillators - Presence of conditions that make it difficult to use a smart device (e.g., moderate/severe dementia, aphasia, etc.) - A person placed under guardianship - Nickel allergy
Secondary users	- Willingness to sign the written informed consent - Lay caregiver helping the involved older adults (family members, helping volunteers) - Professional assistant to their involved older adults (social worker, nurse, physiotherapist, doctor)	- A person without professional experience in the field of aging care
Tertiary users	- Willingness to sign the written informed consent - Persons who are related to home care and social assistance in municipal care, research field, decision-makers in financing and management	- Less than half a year of experience

**Table 2 ijerph-19-16604-t002:** Dimensions and measures of the evaluation study for all the users.

Type of End Users	Dimensions	Scales	Timing of Data Collection		
			T0	T1	T2
Primary users	Health and Wellness Condition	MMSE	X		
		FAC	X		
		Barthel Index	X		
		SF-12v2	X	X	X
		WHO-5 Index	X	X	X
		EQ-5D-5L	X	X	X
	Self-efficacy	GSE	X	X	X
	Usability and Acceptance	SUS		X	X
		UEQ-S		X	X
		QUEST 2.0		X	X
	Privacy and Stigma	Open questions	X	X	X
Secondary users	Usefulness		X	X	X
	Reliability		X	X	X
Tertiary users	Reduction in time		X	X	X
	Reduction in cost		X	X	X
	Reduction in workload		X	X	X

## Data Availability

Not applicable.

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
