# Peer review of "A Technology-Based Intervention to Support Older Adults in Living Independently: Protocol for a Cross-National Feasibility Pilot"

_ijerph, 2022, doi:10.3390/ijerph192416604_

Round 1
Reviewer 1 Report
Your aim is to test the usability and efficiency of the SAVE prototype by systematically and objectively identifying the strengths and weaknesses of the proposed solution for enabling older adults to keep their independent and active lives in their homes and maintain their social relationships for as long as possible.
L106. Describe study design
L132. Data collection was started in September 2022. I think that protocol has been registered and then the study will be started.
L338. Add sample size calculation.
Author Response
Your aim is to test the usability and efficiency of the SAVE prototype by systematically and objectively identifying the strengths and weaknesses of the proposed solution for enabling older adults to keep their independent and active lives in their homes and maintain their social relationships for as long as possible.
Response: Thank you for your time and all your suggestions.
L106. Describe study design
Response: At line 106 we re-phrase as follows “It is a pre-post interventional study involving the use of the SAVE platform for 21 consecutive days. It consists of the use of a mixed-methods approach, in which it collects both qualitative (open questions) and quantitative (standardized tests) data in three different measurements (T0, T1, T2) during the period of use of the system. The data collection card will therefore be divided into three different sections, which correspond to the three different moments of detection: 1) at time 0, before the start of the experimentation (T0); 2) at time 1, after 10 days, i.e., at the midterm of the trial (T1), and 3) at time 2, after 21 days, i.e., at the end of the trial (T2). The users' log data will be continuously stored over the 21 days test period. The research will be managed by qualified personnel. The researchers will ensure the supervision of the tests and the interactions between the users and the system.”
L132. Data collection was started in September 2022. I think that protocol has been registered and then the study will be started.
Response: Thank you for this comment. The registration process of ClinicalTrials.gov admits different recruitment statuses: “not yet recruiting”, “recruiting”, “enrolling by invitation”, “active”, and “completed”. In our case, we registered the protocol after the start of the recruitment.
L338. Add sample size calculation.
Response: Thank you for asking this clarification.
To calculate the sample size, we used G*Power software (latest ver. 3.1.9.4; Heinrich-Heine-Universität Düsseldorf, Düsseldorf, Germany) (Kang, 2021). The G*Power software supports sample size and power calculation for various statistical methods, including dependent t-test (differences between two dependents means - matched pairs). A power calculation based on a dependent t-test shows that at least 34 end-users per group are needed; alpha set at 0.05, beta 0.2 and effect size at 0.5 (medium). Lines: 360-366, p.16.
Moreover, extensive editing of the English language was done.
Reviewer 2 Report
Hello
This is an interesting study that does not raise any significant questions for me. The relevance is determined by the research objectives and population growth.
1. For hardware, I would like to clarify about the limitations that potential users may face?
2. You also specify the manufacturer's company (Galaxy Gear ap), which significantly limits the range of users.
3. Are there plans to expand to other platforms or to other manufacturers
Author Response
Hello
This is an interesting study that does not raise any significant questions for me. The relevance is determined by the research objectives and population growth.
Response: Thank you for your time and all your suggestions.
- For hardware, I would like to clarify about the limitations that potential users may face?
Response: We thank the reviewer for this comment. We changed the title of sub-paragraph 4.1 and added the following sentences from line 410: “Even if we did not expect any negative effects, the presence and use of technological devices (sensors and smartwatch) could be a source of discomfort, anxiety about making mistakes or quickly forgetting the instructions and felt stigma. Indeed, low digital literacy and the poor usability of the devices could hinder older people’s use of technologies. The SAVE technology has been designed and developed according to the UCD approach to counteract these possible limitations by matching the needs and capabilities of users, thereby improving the user experience [27]. The pre-post interventional study design will allow us better to analyse changing behaviours regarding usability, acceptance and stigma.”
- You also specify the manufacturer's company (Galaxy Gear ap), which significantly limits the range of users.
Thank you for this comment. In line 477 you can find our reasons: “Probably the use of specific manufacturer's company (Galaxy Gear app) significantly limits the range of users but the operating systems for smartwatches are still very fragmented and the budget resources of the project are limited. Therefore, for our research, we chose a product of a large company with a significant market share. We aimed for a product that has LTE connectivity (based on eSIM), which makes it independent of a smartphone and allows a better monitoring process. At the time of selection, there were not many products with LTE connectivity, and the eSIM technology was not yet supported by all service providers for all smartwatch producers.”
- Are there plans to expand to other platforms or to other manufacturers.
Thank you for this question and interest in our plans. In line 484 please find our answer: “The architecture we adopted lends itself to easy inclusion of other devices from other manufacturers by adding a software element (an adapter), which is in our expansion plans.”
Round 2
Reviewer 1 Report
Thank you for revision.